# Few-Shot Learning for Medical Image Segmentation Using 3D U-Net and Model-Agnostic Meta-Learning (MAML)

**DOI:** 10.3390/diagnostics14121213

**Published:** 2024-06-07

**Authors:** Aqilah M. Alsaleh, Eid Albalawi, Abdulelah Algosaibi, Salman S. Albakheet, Surbhi Bhatia Khan

**Affiliations:** 1College of Computer Science and Information Technology, King Faisal University, Al Hofuf 400-31982, AlAhsa, Saudi Arabia; ealbalawi@kfu.edu.sa (E.A.); aaalgosaibi@kfu.edu.sa (A.A.); 2Department of Information Technology, AlAhsa Health Cluster, Al Hofuf 3158-36421, AlAhsa, Saudi Arabia; 3Department of Radiology, King Faisal General Hospital, Al Hofuf 36361, AlAhsa, Saudi Arabia; salman049@hotmail.com; 4Department of Data Science, School of Science Engineering and Environment, University of Salford, Manchester M5 4WT, UK; s.khan138@salford.ac.uk; 5Department of Electrical and Computer Engineering, Lebanese American University, Byblos P.O. Box 13-5053, Lebanon

**Keywords:** few-shot learning, MAML, medical image segmentation, meta-learning, U-Net

## Abstract

Deep learning has attained state-of-the-art results in general image segmentation problems; however, it requires a substantial number of annotated images to achieve the desired outcomes. In the medical field, the availability of annotated images is often limited. To address this challenge, few-shot learning techniques have been successfully adapted to rapidly generalize to new tasks with only a few samples, leveraging prior knowledge. In this paper, we employ a gradient-based method known as Model-Agnostic Meta-Learning (MAML) for medical image segmentation. MAML is a meta-learning algorithm that quickly adapts to new tasks by updating a model’s parameters based on a limited set of training samples. Additionally, we use an enhanced 3D U-Net as the foundational network for our models. The enhanced 3D U-Net is a convolutional neural network specifically designed for medical image segmentation. We evaluate our approach on the TotalSegmentator dataset, considering a few annotated images for four tasks: liver, spleen, right kidney, and left kidney. The results demonstrate that our approach facilitates rapid adaptation to new tasks using only a few annotated images. In 10-shot settings, our approach achieved mean dice coefficients of 93.70%, 85.98%, 81.20%, and 89.58% for liver, spleen, right kidney, and left kidney segmentation, respectively. In five-shot sittings, the approach attained mean Dice coefficients of 90.27%, 83.89%, 77.53%, and 87.01% for liver, spleen, right kidney, and left kidney segmentation, respectively. Finally, we assess the effectiveness of our proposed approach on a dataset collected from a local hospital. Employing five-shot sittings, we achieve mean Dice coefficients of 90.62%, 79.86%, 79.87%, and 78.21% for liver, spleen, right kidney, and left kidney segmentation, respectively.

## 1. Introduction

Recent advancements in medical imaging are utilized in healthcare for the diagnosis and treatment of various diseases. They utilize diverse methods and technologies to generate visual depictions of the internal organs and tissues within the human body [1]. Some of those technologies include X-ray, Computed Tomography (CT), Ultrasound, and Magnetic Resonance Imaging (MRI) [2]. Each has its unique benefits and risks [3]. For example, X-rays are known for their low cost and quick acquisition time, which have made them one of the most commonly used technology. Ultrasound imaging is fast, allowing for the real-time visualization of blood flow in arteries using the Doppler shift. MRI produces volumetric images with high spatial resolution, mainly by detecting signals from hydrogen nuclei. This modality uses the abundance of hydrogen nuclei to generate detailed images of internal structures, making it particularly effective for imaging soft tissues. While CT imaging employs X-rays to produce detailed cross-sectional images of the body, it is also widely utilized because of its fast scan time and excellent resolution [1,4].

To facilitate accurate decision-making by physicians, it is crucial to segment key objects and extract essential features from the segmented areas of medical imaging. The segmentation process defines meaningful regions within an image, such as tissues, organs, or pathological conditions. This is important in assisting clinicians in diagnosing diseases, calculating prognoses, and planning medical operations [5]. For instance, segmentation is used to detect infected tumor tissues in medical imaging modalities by separating tumor tissues from normal brain tissues and solid tumors [6].

Segmenting medical images is a demanding and time-consuming task [7]. Detecting abnormalities, particularly rare ones, is time-consuming and costly for medical experts. Consequently, artificial intelligence (AI) has been widely adopted for the automated processing of medical images as a supportive tool for physicians [8]. As a result of the rapid advancement of AI, particularly deep learning (DL), image segmentation methods based on DL have achieved significant success in the field. DL offers many advantages over traditional machine learning and computer vision methods in segmentation accuracy and speed. Using DL to segment medical images can help doctors confirm tumor sizes, quantitatively evaluate treatment effects, and significantly reduce doctors’ workload [9]. While DL has made significant progress and demonstrated high performance in automated general image segmentation, it is also inherently expensive. This is due to the substantial need for annotated images, computing power, and memory resources. Moreover, DL typically demands large datasets, which can be challenging to obtain in the medical domain due to limitations in image availability, privacy concerns, and other issues [8]. Using small datasets may result in overfitting and generalization errors [10]. Additionally, DL struggles to generalize well when applied to unseen datasets [11]. Consequently, few-shot learning is proposed as a solution, enabling the generalization to new tasks with only a few samples. Few-shot learning not only learns rare cases but also reduces the cost and effort associated with data acquisition, along with minimizing processing and computational costs [12].

Recently, few-shot learning has emerged as a promising approach to address data scarcity challenges in the medical field. Few-shot learning models excel at identifying new categories without the need for retraining, making them particularly suited for tasks where labeled data are limited or expensive to collect [13]. Among the various techniques for few-shot learning, one notable method is Model-Agnostic Meta-Learning (MAML) [14]. MAML is an algorithm designed to learn the initialization parameters of a model, allowing for adaptation to new tasks with only a few samples and iterations. It is a versatile few-shot learning algorithm applicable to various types of models and learning problems, including classification, regression, and reinforcement learning. MAML efficiently initializes model parameters, achieving optimal fast learning for new tasks with only a few gradient steps. Furthermore, the MAML technique demonstrates high performance and generalization even on small datasets. Operating in two phases, MAML first undergoes meta-training on a diverse task distribution by iteratively adapting its parameters to new tasks with a limited number of labeled samples. Subsequently, the model is meta-tested on a new, unseen task to evaluate its ability to adapt quickly to novel challenges. MAML has been gaining attention and showing promise in various tasks associated with medical imaging, such as classification [15] and skin lesion segmentation [16].

In this paper, we employ the MAML algorithm in conjunction with a modified 3D U-Net [17] as the baseline network. The synergy between MAML’s capability to learn from small annotated datasets and swiftly adapt to new tasks, combined with the well-established effectiveness of the U-Net architecture in medical image segmentation, results in enhanced segmentation accuracy and efficiency, particularly in scenarios where annotated images are limited.

We have made four key contributions:Utilizing the MAML algorithm with the enhanced 3D U-Net architecture for generalized few-shot medical image segmentation.Evaluating our proposed approach under 5-shot and 10-shot settings.Comparing our proposed approach with existing methods using five-shot settings.Testing the performance of the final models on local hospital data to demonstrate the model’s effectiveness in real-world scenarios.

The remainder of this paper is structured as follows. Section 2 provides an overview of the fundamental and relevant theories. Section 3 highlights previous research studies and identifies gaps. Section 4 describes the adopted methodology. Section 5 presents the experimental results and discussion. Finally, Section 6 presents the conclusion.

## 2. Theoretical Background

This section summarizes the concepts and technologies used in this paper. Section 2.1 focuses on medical image segmentation, and Section 2.2 explains CT modality. Further, Section 2.3 describes the U-Net architecture. Lastly, Section 2.4 discusses Meta-Learning in more detail.

### 2.1. Medical Image Segmentation

Medical image segmentation is fundamental for many medical applications and research purposes, such as diagnosis, treatment planning, and disease monitoring. Its significance lies in its ability to assist radiologists and physicians in making clinical decisions. Additionally, it allows researchers to extract quantitative measurements such as area, shape, intensity, and volume of the segmented regions that provide valuable insights into anatomical structures, disease progression, and treatment planning. As a result, this information is essential for guiding treatment strategies, facilitating early disease detection, and providing insights into how diseases develop.

Medical image segmentation is the process of extracting areas of interest, including organs, vessels, and lesions, from a medical image to emphasize specific information within an image [18]. In other words, the process is about analyzing the medical image by generating a mask to isolate the specific object from the background and unwanted details, where the object has uniform characteristics in texture or gray level. Both types of image segmentation could be applied to medical images. For instance, semantic segmentation is applied in brain tumor segmentation, whereas instance segmentation is applied in cell nuclei segmentation.

Many segmentation software tools are specifically designed for medical imaging, supporting various medical imaging modalities. Medical imaging modalities are generally categorized into structural and functional modalities, depending on the information contained in the image. The modalities include but are not limited to X-Ray radiography, CT, MRI, Positron Emission Tomography (PET), and Single Positron Emission Tomography (SPECT). Figure 1 illustrates the different medical imaging modalities for brain imaging [19]. The software includes various tools and algorithms supporting manual, semi-automatic, or completely automated segmentation. Manual segmentation is time-consuming and labor-intensive and may need expertise. Semi-automatic segmentation techniques combine automated algorithms with user interaction to achieve accurate results. Users often provide an initial region of interest, which guides the algorithm to segment the image as a whole. Manual adjustments, such as refining region boundaries, are also used to decrease segmentation errors. Unlike manual segmentation and semi-automatic approaches that rely on user interactions, fully automatic segmentation methods operate without the need for user interaction. This technique utilizes advanced AI algorithms and computational methods to segment the medical images automatically [20]. DL and U-Net architecture are among the most popular methods for automatic medical image segmentation. Recently, few-shot learning has emerged as a topic of interest and research in medical image segmentation and it is detailed in Section 2.4.1.

### 2.2. Computed Tomography

CT is a type of medical imaging utilizing X-ray technology to generate cross-sectional images of a human body. This technique provides a more comprehensive view of internal structures than traditional X-rays. In the term Computed Tomography, computed means calculation or reconstruction, while tomograph is derived from Greek words as a combination of “tomo” meaning cut or slice, and “graphy” meaning to describe. CT scanner is a specialized machine that typically uses 100 kV to 150 kV [21] of energy to generate images for diagnosing abnormalities and other therapeutic measurements [3]. The basic idea of a CT scanner is to measure the X-ray beam projected through the body by a single detector, and the X-ray tube moves along with the detector [21]. Then, the measurements of the X-rays are processed by computers to create cross-sectional images of the human body called slices. These images show detailed information about the structure and density of various tissues, including bones, organs, blood vessels, and tumors. The computer saves each slice as a file in the Digital Imaging and Communications in Medicine (DICOM) format. Whearse DICOM is the standard format used in medical imaging to store medical images such as CT scans. This file type can contain metadata such as patient information and imaging devices, which are necessary for interpreting and transmitting images [22]. The computer can reconstruct the DICOM files to form a three-dimensional image, allowing healthcare professionals to visualize the body’s internal structures from different perspectives.

The Neuroimaging Informatics Technology Initiative (NIfTI) is a standardized file format that stores and manages medical image data such as CT scans and MRI [23]. NIfTI was adopted as the default format to ensure compatibility and interoperability across different software tools and platforms [24] like 3D Slicer [25] and the Monai framework [26].

### 2.3. U-Net

U-Net is a network architecture primarily designed for medical image segmentation by Ronneberger et al. [27] in 2015. U-Net has performed accurate segmentation for medical image segmentation through pixel-by-pixel prediction. Figure 2 shows the U-Net architecture.

### 2.4. Meta-Learning

#### 2.4.1. Few-Shot Learning

Few-shot learning is a machine learning challenge involving learning from a few samples. It involves a learning task *T* that uses a dataset $D=\{D_{\mathrm{train}},D_{\mathrm{test}}\}$ where $D_{\mathrm{train}}$ is the training set is $D_{train}={\left\{\left(x_{i},y_{i}\right)\right\}}_{i=1}^{I}$, where *I* is a few number of samples, and $D_{test}$ is the testing set $=\{x_{\mathrm{test}}\}$. For input *x* and output *y*, p(x,y) is the joint probability distribution, and ĥ is the optimal hypothesis that maps input *x* to output *y*. Few-shot learning recognizes ĥ by fit $D_{\mathrm{train}}$ and tests on $D_{\mathrm{test}}$. To approximate ĥ, the few-shot learning model defines a hypothesis space *H* of hypotheses h(.;θ)s, in which θ represents each parameter of *h*. A parametric *h* represents a nonparametric model usually requiring massive datasets unsuitable for few-shot learning. Few-shot learning is an optimization technique to search *H* to identify the θ that parameterizes the most suitable $h^{*}$∈H. A loss function assesses the performance of the model denoted as *L*(ĥ,*y*), which is determined over the prediction ĥ = h(x;θ) and the output *y* [12].

#### 2.4.2. Model-Agnostic Meta-Learning

MAML is designed for meta-learning tasks. It utilizes a straightforward task-agnostic algorithm to train model parameters. The objective is to enable the model to learn unseen tasks quickly with just a few gradient updates. MAML was developed in 2017 by Stanford Research and UC Berkeley Alum Dr. Chelsea Finn [14]. The generalized MAML algorithm is explained in Algorithm 1. In MAML, meta-learning refers to a range of tasks p(T) to which the model should be capable of adapting. When the model is trained with K-shot learning, it learns unseen task Ti based on p(T) from *K* samples derived from transition distribution qi and the feedback LTi generated by Ti. The meta-training process involves sampling a task Ti from p(T), training based on *K* samples, updating from the respective loss LTi, and testing the model on unseen samples from Ti. Afterward, the model *f* is enhanced by looking at how the test error changes with parameters to the new data from qi. Then, unseen tasks are chosen from the p(T) during meta-training, and the meta-performance is determined after the *K* samples [14]. The model is depicted by a parameterized function denoted as fθ, where θ corresponds to the model’s parameters. The θ is updated to θi′ when adapting unseen task Ti as shown in Figure 3. Specifically, MAML determines a set of weights θ which can be fine-tuned in an easy way to new testing tasks using the following optimization in Equation (Equation 1) [14]:(1)minθ∑Ti∼p(T)LTi(fθi′)=∑Ti∼p(T)LTi(fθ−α∇θLTi(fθ))
**Algorithm 1** MAML Meta-Lerning**Input:** α: Inner learning rate, β: Outer learning rate, p(T): Variety of tasks**Output:** θi′: Updated set of parameters 1: Randomly initialize θ 2: **while** Training **do** 3:  Select task Ti from p(T) 4:  **for all** Tasks **do** 5:**   for all** samples in Ti
**do** 6:     Evaluate: ∇θLTi(fθ) 7:   **end for** 8:   Calculate adapted parameters: θi′=θ−α∇θLTi(fθ) 9:  **end for**10: Adjust the parameter: θ=θ−β∇θ∑Ti∼p(T)LTi(fθi′)11: **end while**

## 3. Literature Review

This section provides an overview of various techniques used in medical image segmentation and is divided into three sections. Section 3.1 discusses general approaches for medical segmentation, while Section 3.2 introduces some few-shot learning techniques. Section 3.3 discusses different few-shot image segmentation techniques, while Section 3.4 focuses on the application of few-shot learning specifically in the context of medical image segmentation.

### 3.1. Medical Image Segmentation

Many researchers rely on fully convolutional neural networks (FCNNs). Duanmu et al. [29] introduced a brain organ segmentation technique using a 3D FCNN. The method effectively handles significant organ size variation through a weighted loss function and multiple resolution paths, demonstrating good performance in segmenting thin or small organs. Other researchers aim to enhance accuracy while minimizing the size of the training set. Falk et al. [30] implemented a U-Net-based [28] solution for common quantification tasks in biomedical image data, focusing on cell detection and shape measurements. Ibrahim et al. [31] predicted the desired heart mask in MRI using FCNN, introducing both multichannel and single-channel input schemes with U-Net. The multichannel provides more input dimensions, while the single-channel, using a large network, increases network parameters. Yang et al. [32] applied brain tumor segmentation using the basic structure of U-Net, enhancing performance by adding a 1 × 1 convolutional layer for parameter reduction. Various modifications to the U-Net architecture have been proposed in the literature to address challenges such as gradient vanishing and increased complexity. Kerfoot et al. [17] implemented U-Net with residual units [33] on cardiovascular MRI. Hatamizadeh et al. [34] proposed a novel architecture called UNETR, combining transformer architecture with the U-shaped network design for improved capability in capturing long-range spatial dependencies in volumetric medical image segmentation. Jafari et al. [35] introduced DRU-Net, combining the advantages of ResNet and DenseNet on skin lesions and brain images, resulting in fewer hyperparameters, reduced training time, and improved segmentation accuracy. Liu et al. [36] proposed a hybrid approach leveraging pretrained 2D networks for learning features from anisotropic images and then extending it to 3D with anisotropic convolutional blocks. This design allows the effective capture of 3D context while maintaining faster processing times, with skip connections, dense connections, and a pyramid volumetric pooling module contributing to overall performance. Kamiya et al. [37] focused on musculoskeletal analysis by using deep learning to automate the extraction of muscle characteristics from CT images. They introduced 3D recognition of the erector spinae muscle, employing the iterative random forest. Further, they used the FCN-8s network for 2D segmentation of the erector spinae muscle. Additionally, they expanded their scope by applying deep learning to whole-body muscle analysis including segmentation of muscles and bones from whole-body CT images using 2D U-Net. The research also explored the fusion of deep learning with traditional handcrafted feature-based methods to address challenges in preparing labeled training data. The previous networks are specialized for one segmentation task. Inspired by multi-domain learning, Huang et al. [38] presented a multi-domain medical image segmentation approach using a universal neural network architecture that handles multiple tasks. The proposed 3D U^2^-Net aims to learn a universal data representation capable of handling multiple segmentation tasks with a single model. The 3D U^2^-Net consists of separable convolution, which utilizes domain-specific spatial correlations and cross-domain correlations. Their proposed architecture was designed to handle diverse imaging modalities and anatomical structures. However, these segmentation methods heavily rely on CNN. Their effectiveness is predominantly observed in scenarios where a substantial amount of annotated data is available. This scenario changes when confronted with data scarcity in the medical field, where only a limited number of annotated images are accessible.

In semi-supervised learning, most samples are unannotated, with only a few being annotated. On the other hand, unsupervised learning involves scenarios where there are no annotations or human interactions. Due to the limited quantity of annotated images in the medical field, many researchers have turned to unsupervised learning. Two popular methods used in this context are image alignment and feature alignment through adversarial learning. Dog et al. [39] achieved comparable accuracy to supervised learning but used only 10% of annotated brain tumor images. Their segmentation technique was based on a Cycle Generative Adversarial Network (CycleGAN) and used unsupervised domain adaptation and semi-supervised learning. Chen et al. [40] introduced an unsupervised domain adaptation framework that combines image alignment and feature alignment guided by adversarial learning. The authors demonstrated improved segmentation performance on unannotated target images. However, these methods necessitate retraining models using images from the target domain, whether annotated or unannotated. Gathering data from every new target domain for model adaptation is time-consuming and impractical in many real-life scenarios.

Recently, self-supervised learning has gained attention in medical segmentation. In self-supervised learning, data representations are acquired without human annotations. The model is initially trained on unannotated medical images, which are easier to obtain, and later fine-tuned using a small set of annotated medical images. Self-supervised learning involves training embedding functions through self-supervision, aiming to identify more generalizable and transferable features [41]. Various data-augmentation techniques are commonly applied in this approach. Ma et al. [42] utilized adversarial learning to acquire vessel representations based on unannotated images. Adversarial learning is employed to simultaneously generate fake and segmented vessels from unannotated images. This method exhibits significant superiority with performance comparable to unsupervised and traditional techniques. In a different application, Zhang et al. [43] presented a framework for detecting and segmenting cystic lesions in lung CT images. The framework involves two stages: unsupervised segmentation followed by segmentation networks. In the unsupervised segmentation stage, initial annotations for all images are obtained using K-means graph cuts. These annotations are then utilized as labels for a U-Net. Subsequently, a new network is trained recursively from previous predictions. The authors achieved significantly improved segmentation accuracy compared to the segmentation generated using the K-means method. However, since the authors applied unsupervised learning, they needed to manually set the cluster numbers in various applications. Additionally, there could be challenges related to imbalanced label or class distribution. It is worth noting that self-supervised learning is computationally demanding and slower due to the additional annotation task.

### 3.2. Few-Shot Learning

Few-shot learning emulates the learning capability of humans when provided with limited examples. It aims to achieve generalization to related but unseen examples using only a small number of instances, making it particularly valuable in the context of medical images. Researchers have introduced a variety of approaches, encompassing both data-level and meta-learning methods.

#### 3.2.1. Data-Level

These approaches aim to mitigate the overfitting problem associated with small datasets by expanding the dataset through the generation of additional samples. Kumar et al. [44] focused on generative models for text classification and augmented data using various Feature Space Data Augmentation (FDA methods. FDA involves learning a data representation or feature extractor, which is then used to create new data in the feature space. The classifier is subsequently trained on both the original and augmented data. Chen et al. [45] proposed a method that combines Generative Adversarial Networks (GANs) to generate diverse, high-quality medical images across various modalities, including CT scans, X-rays, and pathology images. They evaluated their method on eight datasets to demonstrate the method’s efficacy and generalizability. Li et al. [46] introduced a data-augmentation method based on the Conditional Wasserstein Generative Adversarial Network (cWGAN). Their approach utilizes features from a few labeled samples to construct synthetic features for new classes. It is important to note that the success of data augmentation is contingent on the characteristics of the input dataset. If the input data exhibits biases, the generated data may also inherit those biases. Additionally, for effective data augmentation, a reasonably large training set with sufficient samples is still required.

#### 3.2.2. Meta-Learning

Meta-learning, often referred to as “learning to learn”, aims to enhance the adaptability of automatic learning systems for solving various learning problems. There are three well-known approaches to meta-learning: model-based, metrics-based, and gradient-based meta-learning.

In model-based meta-learning, models are initialized based on known classes and then rapidly adapted to new classes using a small number of examples and gradient updates. Meta-learning models update their parameters after a few training steps, which can be performed independently or controlled by another meta-learner model. Examples of model-based approaches include meta-networks and memory-augmented neural networks. For instance, Santoro et al. [47] employed model-based meta-learning for the one-shot learning problem by utilizing a memory-augmented neural network to retrieve and store memories for each classification task. This method achieved competitive results and demonstrated quick adaptation to new tasks with limited samples. However, it relies on the use of RNN, which can be computationally complex.

In metrics-based meta-learning, the objective is to minimize the distance between feature vectors corresponding to images of the same class while increasing the distance between feature vectors of different classes. The focus is on learning transferable feature embeddings that can generalize from seen to unseen categories. This approach is conceptually similar to nearest neighbor algorithms like k-NN [48] and k-means [49]. For instance, WANG et al. [50] adopted a Siamese network for leaf classification, ensuring that similar samples are close together and dissimilar ones are distant. Li et al. [51] improved matching networks by introducing a category traversal module, which scans all classes in the support set together to identify relevant dimensions for each task. Wu et al. [52] proposed a position-aware relation network based on CNN to learn a suitable distance and relation score, enhancing feature extraction and improving generalization for few-shot problems. Li et al. [53] enhanced prototypical networks by introducing an adaptive margin loss, utilizing a class-relevant additive margin loss to separate samples based on semantic similarity in the feature embedding space. While metrics-based meta-learning is effective and straightforward, its application to tasks like regression or reinforcement learning can be challenging, and it often requires specialized architecture.

In gradient-based meta-learning, the objective is to utilize small support sets to rapidly update the learners’ parameters for effective adaptation to new tasks. The approach involves enhancing the optimization algorithm until the model becomes proficient in learning from a few examples. Several methods have been proposed by different researchers. Santoro et al. [47] incorporated LSTM to obtain valuable data representations through gradient descent. The authors employed a gradually learned abstract method to extract helpful representations from the dataset using gradient descent, along with a mechanism for rapidly incorporating unseen information after one presentation using an external memory module. Zhong et al. [54] introduced a remote sensing image retrieval system built upon MAML. Their approach included three core learning modules: image feature extraction using DNN, mAP optimization using histogram binning, and few-shot learning using MAML. This design allows for effective adaptation to new tasks with only a few samples and iterations.

### 3.3. Few-Shot Image Segmentation

Few-shot image segmentation is an emerging research area that significantly reduces the need for human supervision. It involves predicting areas for new classes based on only a few annotated images. Several studies have explored different approaches in this domain. Wang et al. [55] introduced non-parametric metric learning into few-shot image segmentation. The authors employed a prototype alignment network to extract robust prototypes from the support set. The segmentation of query images is achieved by assigning each pixel a class based on the nearest prototype. Lu et al. [56] guided the segmentation of query images by aligning and matching the features of target classes. The authors also incorporated an LSTM optimization network to enhance predictions iteratively without forgetting the internal segmentation cues. Cao et al. [57] proposed a novel framework called MetaSeg, which combines meta-learning with the supervised learning semantic segmentation methods. MetaSeg can learn an effective initialization and parameter update strategy by distributing few-shot semantic segmentation tasks on meta-training classes.

### 3.4. Few-Shot Medical Image Segmentation

Recently, some researchers have explored few-shot learning for medical image segmentation. Zhao et al. [58], utilizing data augmentation, introduced a one-shot method for MRI brain segmentation. The fundamental concept involves transforming annotated samples and images to generate new annotated samples. However, it is important to note that the segmentation model still requires retraining whenever a new class is added. Ruiwei et al. [59] merged few-shot learning with interactive segmentation to alleviate the annotation burden associated with traditional supervised DL algorithms. The authors introduced multiple branches with robust connections for few-shot medical image segmentation. Additionally, they presented a novel algorithm addressing the limitations of existing few-shot segmentation methods through user interaction. The algorithm demonstrated enhanced performance in trained tasks such as liver and stomach segmentation. However, it may not perform as effectively for other tasks, and the algorithm might incur additional time and cost due to the need for expert involvement. Tomar et al. [60] applied self-supervised learning to acquire random spatial and style representation through a one-shot atlas-based approach utilizing a prior distribution. The authors initially trained the model by employing the style encoder model to learn the similarity between distributions. Subsequently, they applied self-supervised learning to cluster images. Additionally, the authors utilized the appearance model and flow decoder to generate new images and their segmentations. Gama et al. [61] applied MAML to few-shot segmentation tasks, incorporating sparse annotations during meta-training and dense annotations during meta-testing. To address computational constraints, the authors introduced miniUNet, a simplified version of the standard U-Net with three encoder and decoder blocks instead of four. The experiments conducted involved four chest X-ray datasets. This approach enabled the model to predict dense labels by leveraging knowledge acquired from sparse annotations.

In summary, while extensive research has been conducted on image segmentation using DL techniques, their suitability for the medical field is limited due to poor generalization with a small number of samples. To overcome this limitation, few-shot learning has gained widespread application in the medical domain, as it demonstrates strong generalization with only a few samples. MAML, a gradient-based meta-learning technique, offers the ability to quickly adapt to unseen samples with just a few gradient steps. More recently, researchers have integrated U-Net with some few-shot gradient-based algorithms, presenting a promising avenue for further exploration and contribution.

## 4. Methodology

This section provides details on the algorithm, backbone network, and methodology used to obtain the experimental results. Specifically, Section 4.1 introduces the proposed architecture, and Section 4.2 discusses the backbone network.

### 4.1. The Proposed Architecture

In our approach, we have devised a tailored architecture that combines the MAML algorithm with an enhanced 3D U-Net for the segmentation of medical images. While MAML focuses on optimizing the model’s parameters and is compatible with various network architectures, we have specifically chosen to integrate it with the 3D U-Net for improved medical image segmentation. The U-Net architecture stands out as a highly effective network for image segmentation, known for its flexibility, modular design, and success across various medical image modalities [62]. Its versatility extends to handling both 2D and 3D images from different modalities like MRI and CT, making it particularly suitable for a range of medical imaging applications. The enhanced 3D U-Net variant employed in our approach emphasizes 3D medical image segmentation and incorporates residual units. These residual units contribute to faster training and enhance the network’s resilience to inputs that deviate from the training data. The MAML algorithm operates by training a model on diverse tasks, enabling it to quickly adapt to new tasks with only a limited number of training samples. This characteristic aligns well with the challenges posed by medical images, where data scarcity is common. The ability of MAML to fine-tune a model to specific characteristics of each human organ, which can vary significantly, is particularly advantageous for improving segmentation accuracy. Furthermore, MAML facilitates learning a robust initialization that can be fine-tuned for various medical image segmentation tasks, reducing the necessity to train a new model from scratch for each specific task. This not only saves time but also conserves computational resources.

Our proposed architecture trains an enhanced 3D U-Net using the MAML algorithm and follows a comprehensive three-phase approach: meta-training, meta-testing, and final testing. The proposed architecture is depicted in Figure 4. In the meta-training phase (Figure 4a), the MAML algorithm optimizes the model’s parameters by iteratively updating the initial parameters after each epoch until reaching optimal values. During this phase, MAML trains distinct U-Net models on various tasks, each representing a specific segmentation problem (e.g., liver segmentation, spleen segmentation, and right kidney segmentation). For each task, a support set consisting of a few 3D images is utilized to train a model, resulting in different optimal parameters $\theta_{i}$ for each task, as depicted in Figure 5. Subsequently, a query set comprising a few 3D images is used to test each parameter $\theta_{i}$, and the corresponding losses are calculated. The MAML algorithm then backpropagates across the sum of these losses, updating the final parameter θ. This iterative process continues until obtaining the optimal parameters θ that generalize well across all tasks. In the meta-testing phase (Figure 4b), the model is fine-tuned on new tasks, such as left kidney segmentation, using the learned parameter $\theta^{\prime }$ obtained during the meta-training phase. The fine-tuning process involves updating the model’s parameters through a few gradient steps with a small number of 3D images. Importantly, in this phase, the model learns the optimal parameters for the new task leveraging knowledge acquired from other tasks in the meta-learning phase, eliminating the need to start from scratch. Finally, in the final testing phase (Figure 4c), the performance of the final model is evaluated on unseen images belonging to the same tasks used in the meta-testing phase. This phase serves as a crucial evaluation step to assess the model’s generalization capabilities on entirely new but related data.

### 4.2. Backbone Network

The standard U-Net was originally constructed as a fully convolutional network and adapted to function with a reduced set of training images. The 3D U-Net [63] serves as an extension of U-Net specifically designed for the volumetric segmentation of 3D medical images, such as CT or MRI scans. Unlike its predecessor, it employs 3D convolutions to capture features from the entire image volume of the image rather than individual 2D slices.

Our approach adopts an enhanced version of U-Net [17] as the baseline network, leveraging its numerous advantages. These advantages will be elaborated upon in subsequent paragraphs. Additionally, we tailor this network into a 3D U-Net, utilizing 3D convolutions to address the high-dimensional nature of the data.

The enhanced U-Net extends the original U-Net architecture by incorporating layers in the encoding and decoding stages defined using residual units. In the encoding part, each box is labeled with the output volume shape and is implemented with a residual unit. In the decoding part, each box is implemented with an upsampling unit, maintaining similar output volume dimensions and levels as the encoding layer. The utilization of Parametric Rectifying Linear Unit (PReLU) enhances activation learning, resulting in improved segmentation [17]. PReLU is a generalized parametric formulation of ReLU and can be defined as follows [64]:(2)$$f\left(y_{i}\right)=\left\{\begin{array}{cc} y_{i} & \mathrm{if}\;y_{i}>0\;\\ \alpha_{i}y_{i} & \mathrm{if}\;y_{i}<=0 \end{array}\right.$$

The variable $y_{i}$ represents the input to the activation function from the *i* layer, where *i* denotes the number of channels. Each layer learns the same slope parameter, denoted as $\alpha_{i}$. Additionally, the enhanced U-Net incorporates instance normalization to mitigate contrast variations, ensuring that input images are not influenced by varying contrast levels when batched together with images having different contrast ranges. As illustrated in Figure 6, the enhanced U-Net utilizes convolutions with a stride of two for down-sampling and transpose convolutions with a stride of two for up-sampling, instead of using pooling and unpooling layers. These adjustments enable the network to learn the most effective down-sample and up-sample operations while reducing the number of layers within the network’s units [17]. Moreover, we set the input channels to one, signifying that our mask images have only one channel, either zeros or ones. For the output channels, we set it to two, one channel for the background and the other channel for the foreground. In other words, the output channels correspond to the number of classes. Additionally, we set the network’s dropout to 0.1, as the dropout is a widely adopted regularization technique to prevent a network from overfitting during the training phase [65]. Finally, we used batch normalization, a technique used to normalize the activation in the intermediate layers of the network. Batch normalization enhances training speed and accuracy [66].

## 5. Experimental Results and Discussion

This section presents the results and analysis of the experiments, including findings obtained from two datasets under two experimental settings: 10-shot and 5-shot scenarios. Section 5.1 includes an overview of the comprehensive datasets. Furthermore, Section 5.2 presents the experimental settings. Section 5.3 shows the meta-learning results, Section 5.4 discusses the one-way 10-shot results, while Section 5.5 provides a comparative analysis, where the one-way five-shot results are presented and compared against different methods regarding accuracy, time efficiency, and the number of parameters. It also presents the performance of our models by testing them on the hospital dataset. Finally, Section 5.6 provides a further discussion.

### 5.1. Datasets

We used two distinct datasets. The first one is the publicly available TotalSegmentator dataset [67], which is summarized in Section 5.1.1. This dataset played a primary role in both meta-training and meta-testing. The second dataset was obtained from a local hospital and was employed to assess the performance of the final models presented in Section 5.1.2.

#### 5.1.1. TotalSegmentator Dataset

As part of our experiment, we trained our models using the publicly available TotalSegmentator dataset [67], which encompasses a diverse range of medical images depicting various health conditions. This dataset comprises 1204 CT images, with segmentation provided for 104 anatomical structures, including 27 organs, 59 bones, 10 muscles, and 8 vessels. All images are in 3D NIfTI format, exhibiting distinct sizes and slice numbers. Annotations for the images were performed using public AI models whenever available, and subsequent manual refinement of segmentations was conducted as needed. For anatomical structures lacking an AI for automatic segmentation, manual annotation was carried out. The refinement and manual segmentation processes were overseen by board-certified radiologists.

Due to hardware limitations, we narrowed our focus to four specific anatomical structures: the liver, spleen, right kidney, and left kidney. More precisely, we randomly selected 50 3D images, each paired with 200 corresponding anatomical structure masks.

#### 5.1.2. Hospital Dataset

The hospital dataset was collected following the receipt of ethical approval No. 10-EP-2023. The dataset comprises five 3D images obtained from a public hospital in Saudi Arabia. Specifically, Siemens SOMATOM Perspective CT Scanner with a mm slice thickness was used for image acquisition. Table 1 provides information on the gender and age of each patient. Each image was meticulously annotated, encompassing segmentation details for four organs: liver, spleen, right kidney, and left kidney. In total, the dataset consists of five CT images and 20 segmentation images, all in NIfTI format. The annotations were performed using a 3D Slicer by a radiology specialist.

### 5.2. Experimental Setup

#### 5.2.1. Task Generation

The MAML algorithm trains a model using a task distribution p(T). In our approach, we focused on four tasks, each corresponding to the segmentation of a specific organ: the liver, spleen, right kidney, and left kidney. Figure 7 displays these four human organs. We employed N-way k-shot learning in all experiments, where N represents the number of tasks, and k denotes the number of images per task. We conducted four experiments, training four models on three tasks during meta-training, while reserving the last task for meta-testing. This final task remains unseen during meta-training, providing insight into how well the model generalizes to new tasks. During the meta-training phase, each task comprises ten images in a support set and an additional ten in a query set. Furthermore, ten images were used to assess the meta-training models. In meta-testing, we utilized twenty unseen images, with ten or five for training and the remaining ten for testing. Table 2 provides detailed information for each experiment. Throughout all tables and figures, we denote “right” as “R” and “left” as “L” for clarity.

#### 5.2.2. Implementation

The implementation was carried out using a cloud service with GPU support. The server’s CPU was an Intel(R) Xeon(R) CPU @ 2.20 GHz, and the GPU used was the NVIDIA A100-SXM4-40 GB. All experiments were conducted using the Monai [26] framework. Initially, Monai transforms were applied to standardize the intensity range to —200 and 200 and to resize the images to 128×128×256 for the TotalSegmentator dataset and 128×128×128 for the hospital dataset. The network dropout was set to 0.1, and batch normalization was implemented. The network underwent training from scratch using the dice loss until convergence, employing a batch size of 1. An early stopping strategy was applied with a patience parameter set to twenty for meta-training and ten for meta-testing. Parameters were updated using the Adam optimizer, incorporating a learning rate alpha value of 1 × 10^−4^, a beta value of 1 × 10^−6^, and a weight decay of 1 × 10^−5^. DSC served as the primary metric to gauge the performance of all experiments. Additionally, for the final test, other metrics such as the IoU and HD were employed. Table 3 summarizes the hyperparameters.

### 5.3. Meta-Training

For all the experiments presented in Table 2, the meta-training phase yielded promising results, showcasing the models’ capability to acquire generalized knowledge (parameters). This acquired knowledge can be effectively transferred to unseen tasks, leading to a reduction in the required number of images and resources for training. The MAML model’s generalization ability stems from its training on diverse tasks, where each training task encompasses data specific to that particular task. These tasks can be drawn from a distribution of tasks, and the variability introduced is crucial for generalization. It compels the model to capture common patterns and relevant features across a spectrum of tasks. To demonstrate the success of the proposed approach, we evaluated the accuracy of the generalized parameters for each experiment using the DSC. The results are visualized Figure 8.

### 5.4. Results of 1-Way 10-Shot

In this section, we assess our models using the TotalSegmentator dataset, where we set N = 1 and k = 10. Specifically, we conducted the meta-testing, which constitutes the second phase of our experiments, as outlined in Table 2. The mean DSC of our results is 87.62%, the mean IoU is 80.91%, and the mean HD is 12.01 mm. The detailed results are presented in Table 4, highlighting the models’ effectiveness in adapting to new unseen tasks with only ten samples. Random slices for each task are illustrated in Figure 9, showcasing both the Ground Truth (GT) masks and the corresponding predicted masks.

The liver segmentation predictions with a DSC of 93.70% consistently outperform those for other tasks below 90%. This can be attributed to the liver’s relatively fixed, large shape, and distinct boundaries when compared to other organs. In contrast, segmentation for other tasks presents challenges due to the variability in organ shape, size, location, and orientation. To provide a more detailed analysis, we further scrutinized the results by comparing the DSC for each individual image, rather than calculating the average. Figure 10 displays the DSC for each image within every task. Notably, images within a task exhibit normal variations in DSC, influenced by differences in image quality and the level of detail in various features.

### 5.5. Results of 1-Way 5-Shot

In this section, we explore the impact of reducing the support set to half by setting k = 5. Using a smaller value of k offers several advantages, including the ability to train models effectively with relatively small datasets. This proves particularly beneficial in cases involving rare abnormalities and diseases. Moreover, a smaller dataset reduces the burden of data annotation, subsequently lowering annotation costs. The results for the one-way five-shot experiment are presented in Table 5.

When comparing the k = 5 to k = 10 experiments, it is evident that the mean DSC for all tasks decreased by 2.94%. This reduction is anticipated due to the decrease in data volume. Based on the detailed comparison between the two sittings presented in Table 5, all four organs—liver, spleen, right kidney, and left kidney—showed improvement in the DSC when trained on a larger dataset (10-shot) compared to a smaller dataset (5-shot). However, the degree of improvement varied between the organs. For the right kidney, the difference in the DSC was the highest at 3.67%, indicating a significant improvement in segmentation accuracy when using more training data. This suggests that the five-shot model struggled more with right kidney segmentation than other organs. The spleen difference showed a minor improvement compared to other organs with a value of 2.09%, suggesting the model with five shots might have already captured the spleen’s features. The difference in the liver was 3.43% and 2.57% for the left kidney, indicating a moderate improvement compared to the other organs. While all organs benefited from more training data, the right kidney showed the most significant improvement, suggesting it might require a more extensive dataset or more focused training strategies for optimal segmentation accuracy. Overall, the variation in results could be affected by different factors such as anatomical complexity, natural anatomical variability, size and visibility, and data quality.

In the following sub-sections, we expand our results by evaluating the performance of our 5k model on various datasets and comparing it against different medical image segmentation methods.

#### 5.5.1. Comparison with Existing Medical Image Segmentation Methods

We evaluated our method on the TotalSegmentator dataset, targeting the four specified tasks. Our approach demonstrated superior performance compared to other existing medical image segmentation methods, including AHNet [36], Basic U-Net [30], U-Net [17], and UNETR [34], for all tasks. The mean DSC across all tasks reached 84.68%, with an HD of 11.92%. Table 6 and Table 7 present the results of evaluating our approach against these existing medical segmentation methods, utilizing a five-shot scenario and employing various evaluation metrics. In these experiments, AHNet exhibited the lowest DSC of 50.11%, indicating a relatively lower match between predicted and actual segmentation masks. Moving upwards, U-Net alone demonstrated better performance with a DSC of 65.10%. UNETR followed with a DSC of 68.16%, and BasicUNet achieved a DSC of 70.19%, signifying improved segmentation capabilities. However, our approach emerged as the top-performing method, showcasing its ability to learn effectively from limited training data and generalize to new tasks with high accuracy. This is in contrast to the other methods, which may require more annotated data to achieve comparable results. By outperforming U-Net with a mean DSC increase of 19.58%, our approach demonstrates the effectiveness of incorporating the MAML algorithm with the enhanced 3D U-Net. The segmentation results of various methods on the TotalSegmentator dataset are visualized in Figure 11.

#### 5.5.2. Evaluation on the Hospital Dataset

We conducted further evaluations of our models using a real-world dataset, enabling an assessment of their performance and robustness. This evaluation aims to demonstrate the potential utility of our models in real-world applications. Testing the data across all four tasks, we achieved a mean DSC of 81.99%, an IoU of 74.72%, and an HD of 10.88 mm for all tasks combined. Additional details can be found in Table 8, and Figure 12 displays the segmentation result for randomly selected slices. The results highlight a high DSC for the liver task, primarily due to the consistency of the liver among individuals. However, the other tasks exhibited slightly lower DSC, influenced by two factors: first, the model’s lower performance, and second, the inconsistency of the spleen and kidneys among individuals in terms of size, position, and orientation. Figure 13 illustrates an example of the variation in spleen characteristics among different individuals. Despite considering only five images, our results underscore the robustness and reliability of our approach across all tasks.

#### 5.5.3. Robustness to Noise

We trained and evaluated our approach on the TotalSegmentator dataset, which provides a comprehensive and highly varied dataset, including data from different years, resampled to 1.5 mm isotropic resolution, and various levels of ambiguity (e.g., structures highly distorted due to pathology). The dataset contains a wide range of different pathologies, scanners, sequences, and institutions.

For further study of robustness to noise, we applied Gaussian noise to the hospital dataset with a mean of 0.0 and a standard deviation of 0.2, simulating realistic noise situations without significantly reducing image quality. Table 9 shows the results of the noisy images using DSC, IoU, and HD. The results demonstrate that our method maintains a high level of accuracy despite the presence of noise, with only a minor decrease in the mean DSC. For the original images, the mean DSC was 82.14%, while for noisy images, the mean DSC was 82.12%. These results show that our approach can effectively handle noise, ensuring reliable performance in clinical applications where noise is possible. Future work will explore robustness to other types of noise and artifacts, further enhancing the applicability of our method in diverse medical imaging scenarios.

#### 5.5.4. Method Efficiency

Time efficiency is crucial as it directly impacts speed and resource requirements. Table 10 compares the time, while Table 11 compares the number of parameters used for the different methods. Notably, our approach demonstrated improved time efficiency, exhibiting a shorter execution time while maintaining a similar number of parameters as U-Net. This efficiency is attributed to the fact that MAML does not expand the number of learned parameters. These results affirm the effectiveness and efficiency of our approach.

While most of the methods are based on the U-Net architecture, they differ in complexity. This is due to the different architectural designs, including the number of layers, types of layers, and connectivity types. Also, the integration with other architectures may lead to more complexity. For instance, the complexity of UNETR compared with the U-Net arises due to its integration of transformer components, sequence processing mechanisms, and attention mechanisms. While the enhancements have shown a good performance, they necessitate additional computational resources and architectural complexity.

### 5.6. Discussion

By training our models on a diverse set of tasks during the meta-training phase, we achieve a generalized understanding of underlying patterns and structures within medical images. This foundational knowledge, supported by the robust performance of parameters $\theta^{\prime }$ across all tasks (Section 5.3), empowers the models with remarkable adaptability to quickly grasp new tasks even when presented with limited samples during the meta-testing phases (Section 5.4 and Section 5.5). This impressive ability to apply knowledge to unseen tasks, even with limited annotated images, is a testament to the transformative power of generalization in medical image segmentation. Importantly, our models consistently outperformed traditional supervised learning methods, further underscoring the value of this approach. Moreover, successful testing on a hospital dataset validated the models’ cross-domain generalizability, ensuring their applicability to diverse clinical scenarios. The exceptional performance of our models in both 10-shot and 5-shot scenarios resoundingly affirms their effectiveness in resource-constrained settings. These remarkable results, as measured by the DSC, IoU, and HD metrics, demonstrate the models’ exceptional ability to extract maximum learning from minimal annotated images. This accomplishment highlights the transformative potential of few-shot learning in medical image segmentation, indicating a shift in addressing the challenges posed by limited data availability.

Notably, the improved 3D U-Net, when used as a baseline for MAML, achieved impressive results while maintaining a compact model size similar to the baseline-enhanced 3D U-Net. This simple architecture, with its reduced computational requirements, has significant implications for our field, where training models for various 3D medical image tasks is crucial. By minimizing computational overhead, we enable greater accessibility and scalability in medical image analysis.

## 6. Conclusions

In this paper, we integrated the MAML algorithm with U-Net to address the challenge of medical image segmentation in scenarios with limited annotated images. Our approach effectively learned efficient parameters that rapidly adapted to new tasks, yielding satisfactory segmentations. This adaptability allows its application to diverse tasks in the future, including those involving new organs, tumors, and rare diseases, thereby reducing training time and resource requirements. We evaluated our method on the TotalSegmentator dataset in two settings: 10-shot and 5-shot. To demonstrate its effectiveness, we tested our models on a hospital dataset, showcasing their segmentation capabilities on real-world data. In comparison to existing methods, our approach successfully improved the mean DSC by up to 15.91%. Notably, relying solely on U-Net did not yield high results due to the limited number of available images, as U-Net traditionally depends on large datasets for training. The key differentiator of our approach lies in the rapid adaptability and learning capacity of MAML, which enables the effective utilization of limited image datasets. This results in a notable enhancement in accuracy and time efficiency. The learned initialization parameters from the meta-training phase serve as a robust starting point for U-Net, contributing to the overall success of our proposed method.

Training four models on distinct tasks using both datasets underscores the robustness of our approach, particularly in liver segmentation. However, we acknowledge the challenge of segmenting other organs due to their distinctive features, including variable shapes and anatomical variations. Recognizing the pivotal role of preprocessing in enhancing training results for medical images, we propose exploring varied processing techniques tailored to each task. This tailored approach may yield superior segmentation outcomes compared to a one-size-fits-all preprocessing strategy. Specifically, we plan to apply specific intensity normalization and contrast enhancement techniques for each organ separately to better highlight the organs. Additionally, we plan to use advanced filtering techniques such as anisotropic diffusion to reduce noise and preserve edges and fine details. Moreover, we will explore the adaptive histogram equalization to improve the visibility of each organ. Therefore, we will conduct comparative studies to evaluate the impact of each preprocessing technique on segmentation performance for each organ, using various metrics, including DSC, IoU, and HD. To further enrich this work, it would be valuable to address, and conduct comprehensive accuracy assessments and comparative studies for specific challenges of data scarcity and investigate clinical cases such as specific tumors or tuberculosis caverns in the lung. This area of research warrants detailed exploration and could significantly enhance the applicability and robustness of the proposed approach. Moreover, it is worth investigating Basic U-Net or UNETR as potential baseline networks for MAML, given their demonstrated competitive DSC. However, this consideration comes with the caveat of increased resource requirements due to the higher number of parameters involved. As part of our future endeavors, we plan to augment the hospital dataset by incorporating images of unhealthy organs. This expansion aims to create a more comprehensive dataset for our research.

## Figures and Tables

**Figure 1 diagnostics-14-01213-f001:**
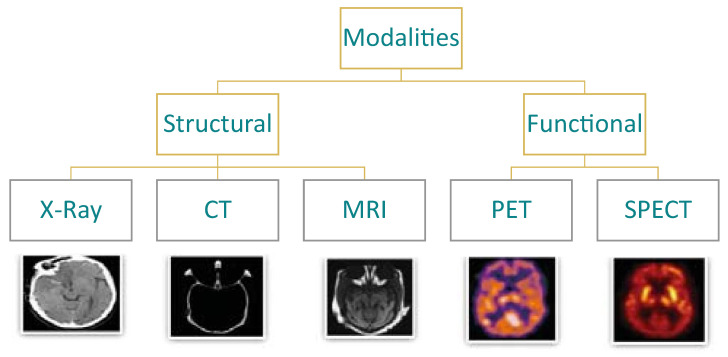
Different medical imaging modalities [19].

**Figure 2 diagnostics-14-01213-f002:**
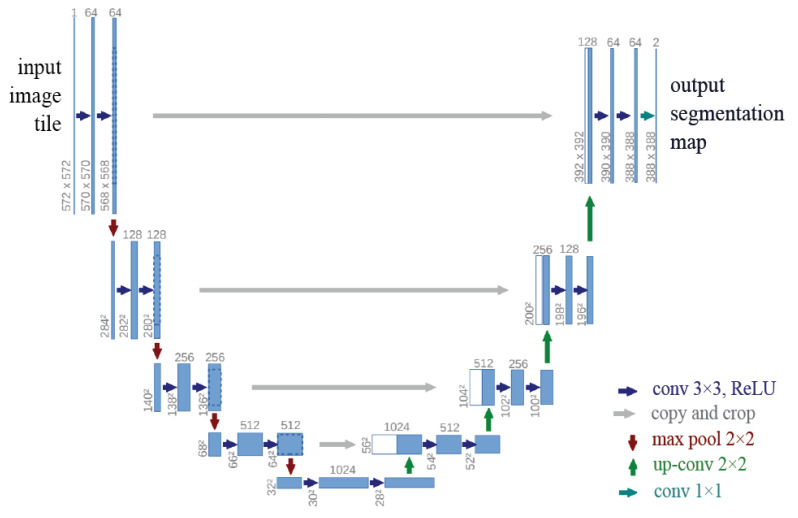
U-Net architecture. The blue boxes refer to a multi-channel feature map. Each box is numbered with the channel number. Feature maps are displayed in white boxes. Arrows indicate various functions [28].

**Figure 3 diagnostics-14-01213-f003:**
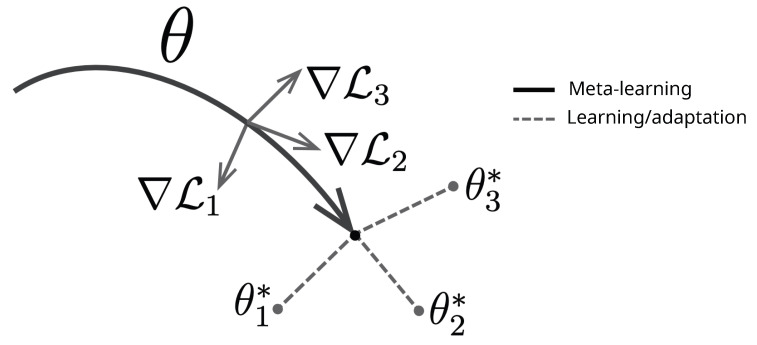
Meta-learning in MAML algorithm [14].

**Figure 4 diagnostics-14-01213-f004:**
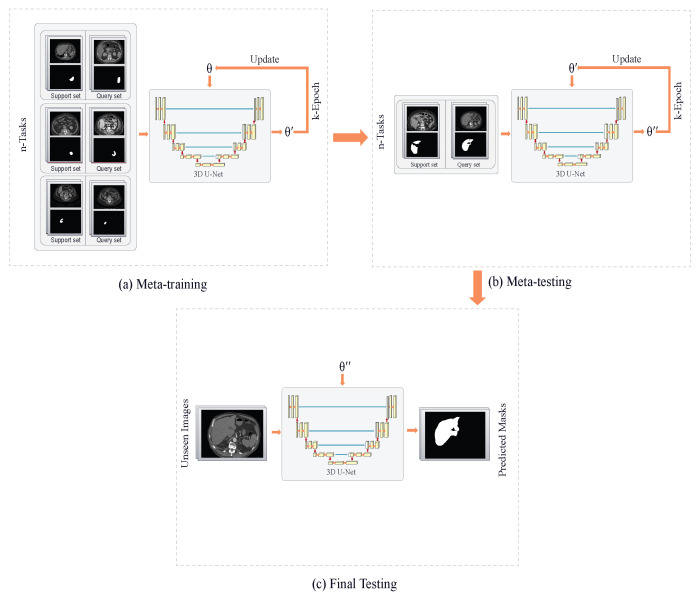
Architecture of our approach. (**a**) Meta-training stage: the model optimizes its initial random parameters θ for different tasks. (**b**) Meta-testing stage: the model uses the generalized parameters $\theta^{\prime }$ and fine-tunes them for the new task. (**c**) Testing stage: the model is evaluated on an unseen dataset but belongs to the same testing task using the fine-tuned parameters $\theta^{\prime \prime }$.

**Figure 5 diagnostics-14-01213-f005:**
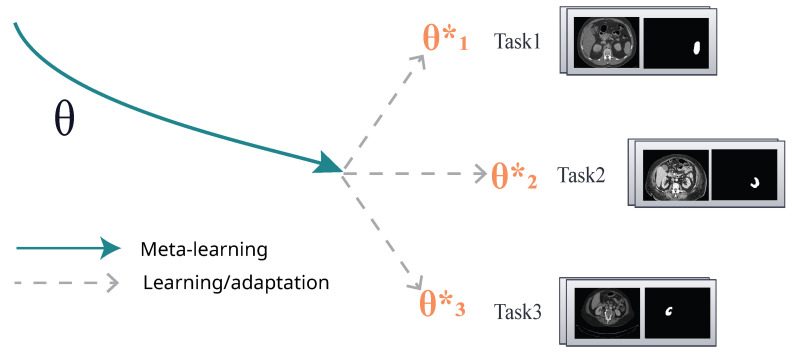
Meta-learning in MAML algorithm.

**Figure 6 diagnostics-14-01213-f006:**
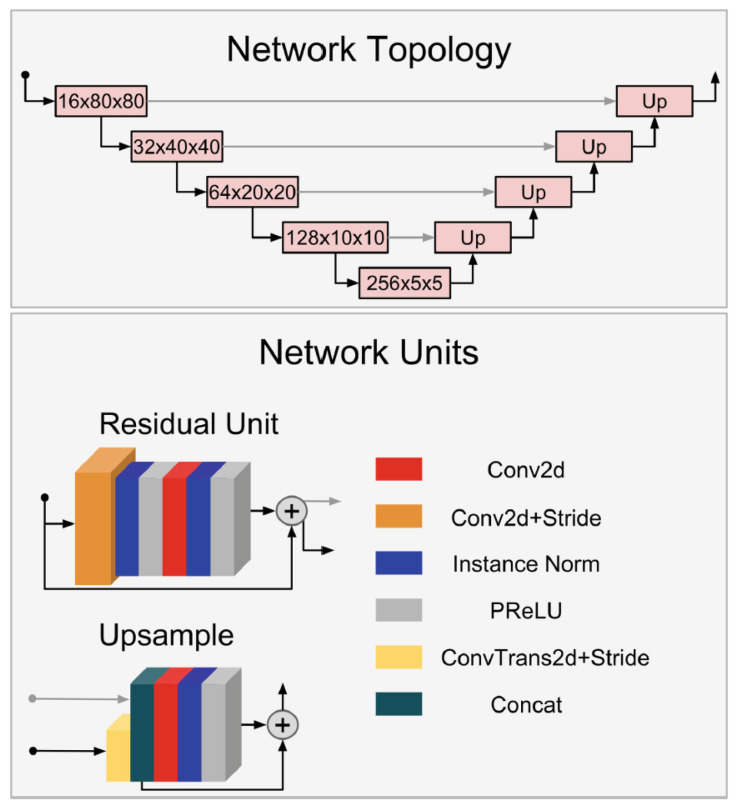
The enhanced U-Net architecture [17].

**Figure 7 diagnostics-14-01213-f007:**
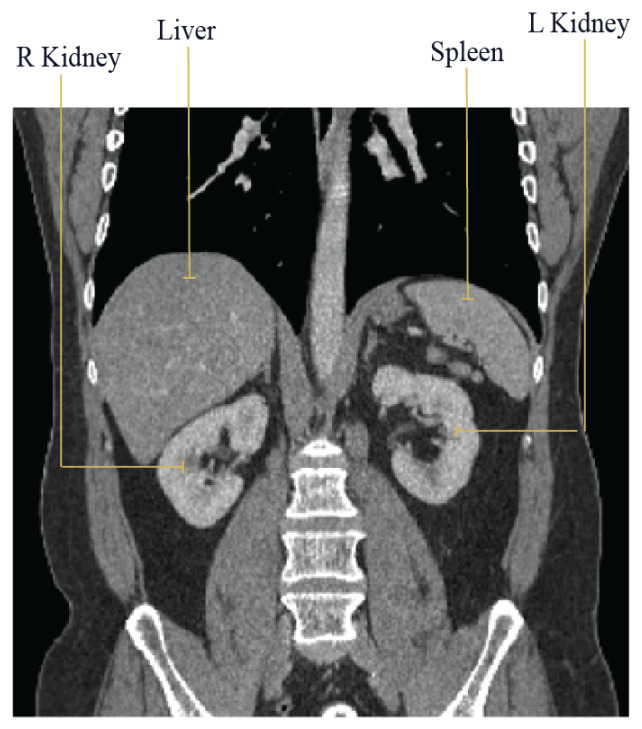
Internal organs of the human body.

**Figure 8 diagnostics-14-01213-f008:**
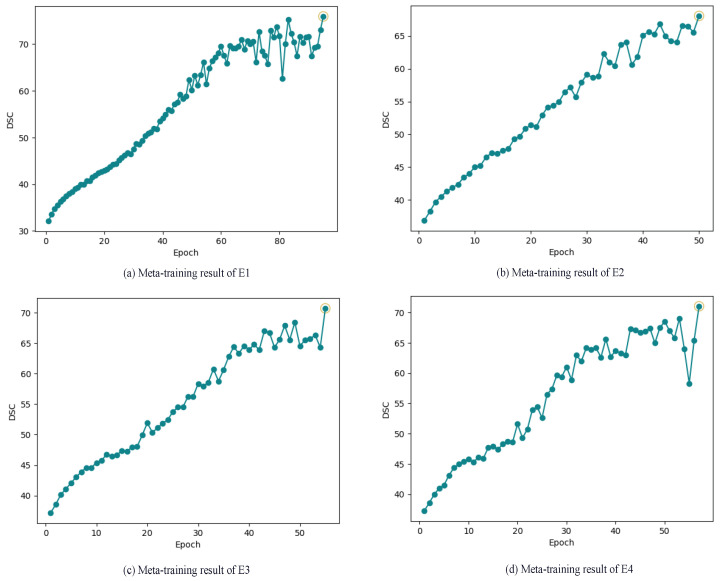
Meta-learning result: DSC (%) VS epochs of all experiments $E_{i}$. (**a**) Meta-training result of E1, (**b**) Meta-training result of E2, (**c**) Meta-training result of E3, (**d**) Meta-training result of E4.

**Figure 9 diagnostics-14-01213-f009:**
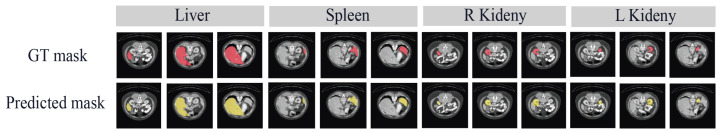
Segmentation results on three slices for each task using 10-shot.

**Figure 10 diagnostics-14-01213-f010:**
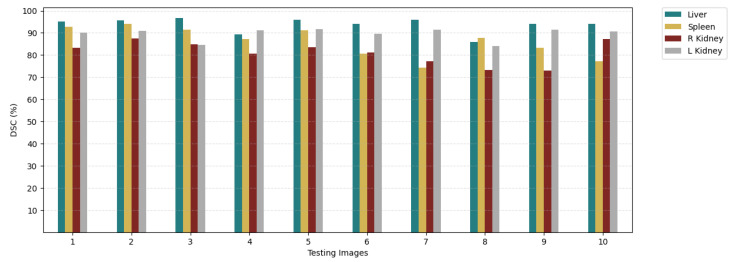
DSC results for each image in each task using 10-shot.

**Figure 11 diagnostics-14-01213-f011:**
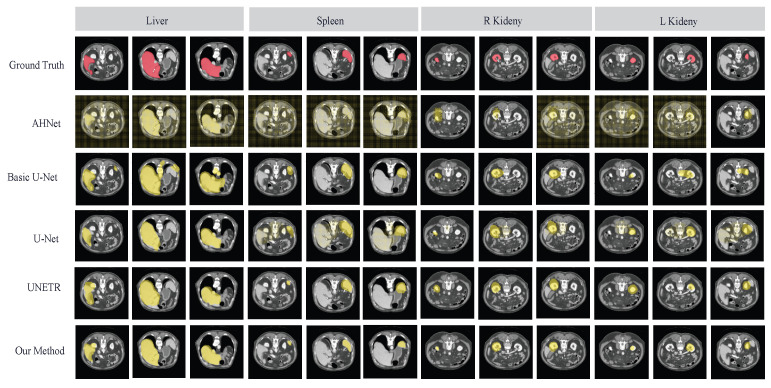
Segmentation results using different methods. The results show three slices for each task.

**Figure 12 diagnostics-14-01213-f012:**
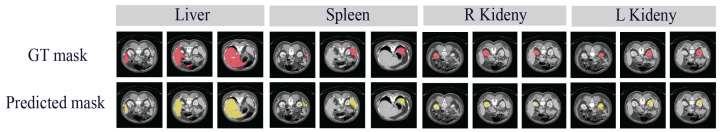
The segmentation results obtained by our approach when applied to the hospital dataset.

**Figure 13 diagnostics-14-01213-f013:**
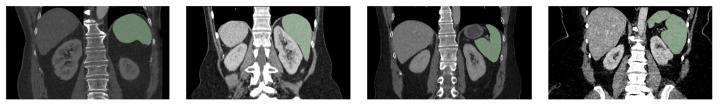
Variation in spleen characteristics among individuals.

**Table 1 diagnostics-14-01213-t001:** Details of hospital dataset.

Patient Number	Gender	Age
1	Male	41
2	Female	29
3	Male	27
4	Male	32
5	Female	38

**Table 2 diagnostics-14-01213-t002:** Number of 3D images utilized per task during both meta-training and meta-testing in each Experiment E.

E	Meta-Training						Meta-Testing		
	Task1	Task2	Task3	Support-Set	Query-Set	Testing	Task4	Training	Testing
1	Spleen	L Kidney	R Kidney	10	10	10	Liver	10/5	10
2	Liver	L Kidney	R Kidney	10	10	10	Spleen	10/5	10
3	Liver	Spleen	L Kidney	10	10	10	R Kidney	10/5	10
4	Liver	Spleen	R Kidney	10	10	10	L Kidney	10/5	10

**Table 3 diagnostics-14-01213-t003:** Hyperparameters used in our experiments.

Parameter	Value
Batch size	1
Dropout	0.1
Optimizer	Adam
α	1 × 10^−4^
β	1 × 10^−6^
Weight decay	1 × 10^−5^

**Table 4 diagnostics-14-01213-t004:** Experimental results on all tasks using 10-shot and DSC (%), IoU (%), and 95% HD (mm).

Method	Liver	Spleen	R Kidney	L Kidney	Mean
DSC	93.70	85.98	81.20	89.58	87.62
IoU	88.95	78.80	72.99	82.89	80.91
HD	20.55	10.03	14.53	2.92	12.01

**Table 5 diagnostics-14-01213-t005:** The difference in DSC (%) between using 10-shot and 5-shot.

k	Liver	Spleen	R Kidney	L Kidney	Mean
10	93.70	85.98	81.20	89.58	87.64
5	90.27	83.89	77.53	87.01	84.68
	3.43	2.09	3.67	2.57	2.94

**Table 6 diagnostics-14-01213-t006:** Experimental results against existing medical segmentation methods on all tasks using 5-shot and DSC (%).

Method	Liver	Spleen	R Kidney	L Kidney	Mean
AHNet [36]	58.96	47.51	47.02	46.93	50.11
Basic U-Net [30]	76.21	68.08	65.38	71.10	70.19
U-Net [17]	86.40	67.76	52.24	53.99	65.10
UNETR [34]	88.29	63.23	57.69	63.42	68.16
**Our Approach**	**90.27**	**83.89**	**77.53**	**87.01**	**84.68**

**Table 7 diagnostics-14-01213-t007:** Experimental results against existing medical segmentation methods on all tasks using 5-shot and 95% HD (mm).

Method	Liver	Spleen	R Kidney	L Kidney	Mean
AHNet	75.36	91.76	87.58	84.75	84.86
Basic U-Net	57.48	66.86	64.75	43.66	58.19
U-Net	31.04	88.42	80.41	78.38	69.56
UNETR	36.84	63.03	70.21	59.74	57.45
**Our Approach**	**15.01**	**07.84**	**19.14**	**05.70**	**11.92**

**Table 8 diagnostics-14-01213-t008:** Experimental results using hospital dataset and DSC (%), IoU (%), and 95% HD (mm).

Metric	Liver	Spleen	R Kidney	L Kidney	Mean
DSC	90.62	79.86	79.87	78.21	82.14
IoU	84.18	72.45	71.75	71.27	74.91
HD	7.77	6.84	14.85	9.88	9.84

**Table 9 diagnostics-14-01213-t009:** Experimental results using a noisy hospital dataset and DSC (%), IoU (%), and 95% HD (mm).

Metric	Liver	Spleen	R Kidney	L Kidney	Mean
DSC	90.66	79.71	79.92	78.41	82.12
IoU	84.23	72.36	71.79	71.44	74.96
HD	7.67	6.86	11.48	9.74	8.94

**Table 10 diagnostics-14-01213-t010:** Comparison of the time efficiency among different methods, with time measured in seconds (S).

Method	Liver	Spleen	R Kidney	L Kidney	Mean
AHNet	184.3	176.2	175	232.5	192
Basic U-Net	46.9	50.2	63.4	48.4	52.225
U-Net	58.4	111.8	79.2	94.1	85.875
UNETR	251.4	189.3	183.6	214.8	209.775
**Our Approach**	**18.2**	**21.8**	**10.1**	**22.7**	**18.2**

**Table 11 diagnostics-14-01213-t011:** Comparison of the number of parameters among different methods.

Method	Number of Parameters
AHNet	38,134,944
Basic U-Net	5,749,410
**U-Net**	**4,808,917**
UNETR	93,404,386
**Our Method**	**4,808,917**

## Data Availability

The data presented in this research are available on request from the corresponding author due to privacy.

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
