# Peer review of "Few-Shot Learning for Medical Image Segmentation Using 3D U-Net and Model-Agnostic Meta-Learning (MAML)"

_diagnostics, 2024, doi:10.3390/diagnostics14121213_

Round 1
Reviewer 1 Report
Comments and Suggestions for Authors
Few-shot learning for medical image segmentation employing U-Net has been proposed.
I have some comments that need to be addressed.
(1) What is the technical contribution of the paper it is well-known that the Few-shot Learning approach using U-Net can provide high performance for segmentation tasks.
(2) Similar work can be seen in recently published articles and integrating MAML has been used which is a strong practice, particularly for few-shot learning tasks, where a model needs to generalize well from only a few examples.
''Chen, Xiaocong, et al. "Generative adversarial u-net for domain-free few-shot medical diagnosis." Pattern Recognition Letters 157 (2022): 112-118.
''Huang, Chao, et al. "3D U 2-Net: A 3D universal U-Net for multi-domain medical image segmentation." International Conference on Medical Image Computing and Computer-Assisted Intervention. Cham: Springer International Publishing, 2019.''
(3) While using MAML did you consider the training time issue as hyperparameter searches to stabilize training and achieve high generalization and being very computationally expensive at both training and inference times?
(4) Extra sections such as 2.3, 3.1.1, 3.1.2, and 3.1.3. These can be reduced to a general sentence.
(5) The quality of the figures such as Fig. 4 and Fig. 10 can be improved.
Comments on the Quality of English LanguageMinor English Proofreading is required.
Author Response
Thank you very much for taking the time to review this manuscript. Your comments and feedback are invaluable to our work.
Please see the attachment.

Reviewer 2 Report
Comments and Suggestions for Authors
This paper presents a method that integrates the MAML algorithm with a 3D U-Net to address the challenge of medical image segmentation with limited annotated images. The proposed approach is evaluated using the TotalSegmentator dataset and a hospital dataset, achieving high accuracy in both 10-shot and 5-shot settings. Moreover, the proposed method successfully improves the mean DSC significantly compared to existing methods.
The main contributions of this research are as follows:
- Utilization of the MAML algorithm with an enhanced 3D U-Net architecture for generalized few-shot medical image segmentation.
- Evaluation of the proposed approach under 5-shot and 10-shot settings.
- Comparison of the proposed approach with existing methods using 5-shot settings.
- Testing the performance of the final models on local hospital data to demonstrate the model's effectiveness in real-world scenarios.
This paper presents an innovative solution to the challenge of medical image segmentation with limited annotated images and makes a significant contribution to the field of medical image analysis. The proposed method demonstrates excellent performance in terms of accuracy, adaptability, and efficiency, showing promise for clinical applications.
However, the following points require further improvement and discussion:
- Validation of the proposed method's generalizability: The current study focuses on a limited set of organs, namely the liver, spleen, and kidneys. However, the discussion on the applicability of the proposed method to other organs, tumors, and rare diseases is insufficient. Furthermore, the TotalSegmentator dataset used in this study contains skeletal muscle regions in addition to the targeted organ regions. The authors should cite a paper on skeletal muscle segmentation (DOI: 10.1007/978-3-030-33128-3_11) and clarify the limitations of their proposed method. Future validation should include a wider range of anatomical structures and medical imaging modalities.
- Consideration of resource requirements: The proposed method achieves a significant reduction in execution time while maintaining a comparable number of parameters to U-Net. However, when using Basic U-Net or UNETR as baseline networks, the increase in the number of parameters may lead to higher resource requirements. This point requires a more detailed discussion.
- Optimization of preprocessing techniques: The authors propose exploring preprocessing techniques tailored to each task to enhance the training results for medical images. However, the specific methods and expected effects are not sufficiently discussed. A more detailed investigation is needed to determine how the optimization of preprocessing techniques influences the performance of the proposed method.
Considering the above points, this paper is a valuable research contribution to the field of medical image segmentation. However, addressing the aforementioned issues and conducting further improvements and validations would enhance the completeness of the paper. A revised version of the manuscript is encouraged for resubmission.
Author Response

(The authors gave the same response as above.)

Reviewer 3 Report
Comments and Suggestions for Authors
The authors proposed a quite interesting algorithm of medical image segmentation, the main benefit ща цршср is possibility to be trained on a very small dataset. But the presentation of the material is more suitable for specialists of image analysis or computer science not specialists who works in the field of medical diagnostics. The scope of the journal “machine learning and artificial intelligence in diagnostics” is not also directly related to this manuscript.
I would like to give some suggestions, which can help the authors to modify this manuscript to a form more acceptable for the Diagnostics.
1. The introduction section
- The introduction is better to begin with a short description of routine medical image methods, their characteristics (like penetration, spatial resolution, contrast, etc.), advantages/disadvantages, limitations. By the way, ultrasound technique did not mentioned in the text.
- After that, the main problem of medical diagnostics based on image analysis can be formulated. For example, the vital problem can be automatic medical image segmentation. The state of art in this field should be included in the text.
- The potential of applications of AI methods like ANNs in this field and examples of their implementation. Here, the authors can formulate the more specific problem of necessity of developing methods of automatic medical image segmentation, which can trained on a very small dataset.
2. The current sections 2.5 and 3 is better to present in Appendix section.
3. In the section 5, it will be very good:
- to consider a clinical case with medical diagnostic decision based on image analysis, for example, detection of tuberculosis caverns in the lung on CT image.
- to estimate accuracy of this case diagnostics using the proposed by the authors method comparing to another reference method.
Additional comments
4. The robustness of the method proposed by the authors regarding the noise on images should be studied and discussed.
5. English should be improved. Examples:
- Line 98. Many segmentation software is built specifically
- Figure 3 legend: Learning/adoaptation
- Figure 5 legend: Adoptation
Comments on the Quality of English Language
5. English must be improved. Examples:
- Line 98. Many segmentation software is built specifically
- Figure 3 legend: Learning/adoaptation
- Figure 5 legend: Adoptation
Author Response

(The authors gave the same response as above.)

Round 2
Reviewer 1 Report
Comments and Suggestions for Authors
I have no further comments as the authors have addressed all my concerns.
Reviewer 2 Report
Comments and Suggestions for Authors
Although the revisions are not perfect, the author has demonstrated a genuine attempt to incorporate the feedback from my review. With this in mind, I support the acceptance of this paper.
Reviewer 3 Report
Comments and Suggestions for Authors
Dear Authors,
This manuscript is quite good but any diagnostics task are not solved. The segmentation is an important step in medical image processing but it alone means nothing for real medical diagnostics decision. I advice you to send this manuscript in a more suitable journal.